# The Type of Fat in the Diet Influences Regulatory Aminopeptidases of the Renin-Angiotensin System and Stress in the Hypothalamic-Pituitary-Adrenal Axis in Adult Wistar Rats

**DOI:** 10.3390/nu13113939

**Published:** 2021-11-04

**Authors:** Germán Domínguez-Vías, Ana Belén Segarra, Manuel Ramírez-Sánchez, Isabel Prieto

**Affiliations:** 1Unit of Physiology, Department of Health Sciences, University of Jaén, Las Lagunillas, 23071 Jaén, Spain; asegarra@ujaen.es (A.B.S.); msanchez@ujaen.es (M.R.-S.); 2Department of Physiology, Faculty of Health Sciences, Ceuta, University of Granada, 18071 Granada, Spain

**Keywords:** aminopeptidase, angiotensinase, dipeptidyl peptidase IV, high-fat diet, hypothalamus-pituitary-adrenal axis, renin-angiotensin system, stress

## Abstract

(1) Background: Prolonged feeding with a high-fat diet (HFD) acts as a stressor by activating the functions of the hypothalamic-pituitary-adrenal gland (HPA) stress axis, accompanied of hypertension by inducing the renin-angiotensin-aldosterone system. Angiotensinases enzymes are regulatory aminopeptidases of angiotensin metabolism, which together with the dipeptidyl peptidase IV (DPP-IV), pyroglutamyl- and tyrosyl-aminopeptidase (pGluAP, TyrAP), participate in cognitive, stress, metabolic and cardiovascular functions. These functions appear to be modulated by the type of fat used in the diet. (2) Methods: To analyze a possible coordinated response of aminopeptidases, their activities were simultaneously determined in the hypothalamus, adenohypophysis and adrenal gland of adult male rats fed diets enriched with monounsaturated (standard diet (S diet) supplemented with 20% virgin olive oil; VOO diet) or saturated fatty acids (diet S supplemented with 20% butter and 0.1% cholesterol; Bch diet). Aminopeptidase activities were measured by fluorimetry using 2-Naphthylamine as substrates. (3) Results: the hypothalamus did not show differences in any of the experimental diets. In the pituitary, the Bch diet stimulated the renin-angiotensin system (RAS) by increasing certain angiotensinase activities (alanyl-, arginyl- and cystinyl-aminopeptidase) with respect to the S and VOO diets. DPP-IV activity was increased with the Bch diet, and TyrAP activity decrease with the VOO diet, having both a crucial role on stress and eating behavior. In the adrenal gland, both HFDs showed an increase in angiotensinase aspartyl-aminopeptidase. The interrelation of angiotensinases activities in the tissues were depending on the type of diet. In addition, correlations were shown between angiotensinases and aminopeptidases that regulate stress and eating behavior. (4) Conclusions: Taken together, these results support that the source of fat in the diet affects several peptidases activities in the HPA axis, which could be related to alterations in RAS, stress and feeding behavior.

## 1. Introduction

Continuous exposure to environmental stressors stimulates a grouped adaptive response, arbitrated by the activation of different neural circuits involved in emotional and cognitive processing, including the hypothalamic-pituitary-adrenal (HPA) axis and associated with the renin-angiotensin-aldosterone (RAAS) system and hypertension [1,2].

Hyperactivity of the HPA axis has been observed in certain subgroups of patients with anxiety and mood disorders [1]. HPA axis is an important neuroendocrine stress system and high fat diets (HFD) are capable of inducing alterations in the basal regulatory activity of the HPA axis, being related to obesity, different responses to the development of metabolic syndrome (MetS), stress, anxiety, anhedonia and certain behavioral changes [3,4,5,6,7,8,9]. The results of the activation of the HPA axis are limited and inconclusive as they differ due to the wide range of animal models and methodological differences used, such as the experimental conditions and the content/duration of the diets [4,10]. The mineralocorticoid aldosterone is an important regulator of blood pressure (BP) and electrolyte balance, being stimulated by HFD and its excess is counterproductive because it is associated with cardiometabolic alterations [11,12,13,14]. A mechanism of adaptation to stress is provided by glucocorticoid hormones, however, variations in their receptor function alter the activity of the HPA axis as compensatory activity, thereby presenting normal lipid and glucose homeostasis [15].

The renin-angiotensin system (RAS) has a fundamental role in cardiovascular physiology through its effects on the regulation of blood pressure and electrolyte balance [16]. On the other hand, also as another adaptation mechanism to the excess of glucocorticoids, it is associated with hypertension due to activation of the RAAS [15,17], and propensity to suffer metabolic diseases via RAAS [2,15]. The obesogenic HFDs appear to be limiting in the adaptations of the activity of the HPA axis and of the RAS [15]. The HPA axis participates in the homeostasis of cardiovascular functions and in the regulation of BP, where the RAS plays an important role [18,19]. The HPA axis and RAAS are well known to be associated with hypertension [20], however, the underlying mechanisms are not yet clear. Angiotensin II (Ang-II), which is associated with hypertension, also induces inflammation of the brain and can induce depressive behaviors through microglial activation in the hippocampus and hyperactivation of the HPA axis in mice [21]. The metabolism of angiotensins and vasopressin depends on the action of certain aminopeptidases (APs) whose activity may be influenced by sympathetic tone [18,19,22]. HFDs influence the classical main components of the local and central RAS, as well as the behavior of other non-classical RAS elements such as Ang 2–10, Ang-III, Ang-IV. These metabolites are regulated by angiotensinase enzymes (Figure 1), among them we find the activities: aspartyl-AP (AspAP; EC 3.4.11.21), hydrolyzes Ang-I to form Ang 2–10; glutamyl-AP (GluAP or AP-A; EC 3.4.11.7), hydrolyzes Ang-II to form Ang-III; alanyl-AP (AlaAP or AP-M; EC 3.4.11.2) and arginyl-AP (ArgAP or AP-B; EC 3.4.11.6), in charge of transforming Ang-III into Ang-IV and Ang 4–8; cystinyl-AP or insulin-regulated AP (CysAP or IRAP; EC 3.4.11.3), which is recognized as the binding site for the Ang-IV receptor (AT4), which in turn shares catalytic centers for oxytocin and vasopressin [22,23,24,25,26,27]. Angiotensinases are known to be closely related to cognitive, metabolic and cardiovascular functions, but the type of fat in the diet can influence angiotensinases and, consequently, behavior [28]. RAS activation stimulates catecholamine and NPY secretion [29,30,31,32], furthermore, NPY itself is also capable of regulating catecholamine biosynthesis in the brain [33] and in the adrenal gland [34,35].

Several studies consider that a diet high in saturated fat, regardless of obesity, alters the function of the HPA axis and contributes in different ways to behavior, stress and anxiety [36,37]. Consuming saturated fat increases sensitivity to NPY [38]. It is suggested that the increase in central levels of NPY mitigate the threshold of response to stress and depression, causing a phenotype resistant to stress in animal models that present mutations in the gene that expresses DPP-IV [39,40]. Dipeptidyl peptidase IV (DPP-IV, CD26; EC 3.4.14.5) is an enzyme associated with the distribution of body fat [41] and modifiable by the type of fat [23,24,42], with post-proline activity that cleaves several relevant peptides for energy homeostasis [23,43], but neuropeptide Y (NPY) is the best and main substrate for DPP-IV [44], a peptide known for its endogenous anxiolytic effect on the central nervous system (CNS) (Figure 1).

The role of different APs that act on substrates that determine the activation of the HPA axis is also recognized, among them: (I) Pyroglutamyl-aminopeptidase (pGluAP; EC 3.4.11.8), an enzyme that degrades the Thyrotropin-releasing hormone (TRH) substrate (Figure 1) [45], implicated in energy metabolism associated with the regulation of eating behavior [23,46] and mental stress disorders [47]. Furthermore, it is known that HFDs have the ability to modify the biological functions of pGluAP [22,23,48]. (II) the tyrosyl-aminopeptidase amino-enkephalinase (TyrAP; EC 3.4.11.-) [49], preferentially releasing amino (NH2)-terminal tyrosine residues [22], absolutely necessary for neuropeptides to exert analgesic effects and as a treatment in the delay of neurodegenerative diseases and mental disorders [50,51], with recognized regulation of intake-satiety, hedonic control, and intermediate metabolism [23].

However, on the HPA axis, the behavior of the dietary fatty acid type on the regulatory mechanisms of the non-classical RAS pathway and its implication as a stress pathway is still not entirely clear. The objective of this study was to investigate the effect of diets high in saturated fat (SAFA) and monounsaturated (MUFA) on the activation of the HPA axis, especially in the relationship between RAS classical/non-classical homeostasis regulatory APs and APs related to stress and energy metabolism under long-term HFD-induced stress conditions. The hypothesis of this work is based on a possible capacity of olive oil to normalize the RAS activities and the activities involved in the regulation of stress and energy metabolism of the HPA axis. This implies a beneficial effect of diets with a higher MUFA content, compared to SAFA diets, by reducing overstimulation of the RAS pathway and its implication in the onset of stress.

## 2. Materials and Methods

### 2.1. Animals and Diets

For the experiments, they were started with adult male Wistar rats from the supplier Harlan Interfauna Ibérica SA (Barcelona, Spain), when they were 6 months old to simulate the effect of a high-fat diet in adults and an approximate body weight of ±495 g, as described [24]. The experimental designs were previously approved by the Institutional Committee for Animal Care and Use of the University of Jaén, with project code number PIUJA_2005_acción 14 (1 January 2006), under Directive 2010/63/EU of the Council of Communities European Regulations and the Spanish Regulation RD 53/2013. The availability of food and drink was ad libitum for 24 weeks, under controlled conditions of 12 h light: 12 h darkness, stable values of temperature (20–25 °C) and humidity (50 ± 5%). The rats were randomly separated by dietary groups: (1) a control group, called the standard diet (diet S, *n* = 6), which corresponded to a commercial feed for laboratory rodents (Panlab, Barcelona, Spain), with a composition nutritional consisting of proteins (16.5%), total fat (3%), carbohydrates (60%, nitrogen-free extract (NFE)), minerals (5%) and fiber (4%). (2) The other two groups corresponded to high-fat diets (HFD), based on diet S but supplemented with a predominantly high composition in MUFA (20% virgin olive oil; VOO diet, *n* = 5) or SAFA (20% butter + 0.1% cholesterol; diet Bch, *n* = 5) to simulate the average cholesterol content of the western diet. The VOO diet contained virgin olive oil from the Cooperativa de los Villares (Jaén, Spain), with a total content (%) of monounsaturated fatty acid (MUFA: 75.5% oleic acid, C18:1 ω-9), acid saturated fatty acid (SAFA: 11.5% palmitic acid, C16:0) and polyunsaturated fatty acid (PUFA: 7.5% linoleic acid, C18:2 ω-6). Furthermore, the VOO diet contained polyphenols (0.105 g/100 g) and α-tocopherol (0.050 g/100 g) as unsaponifiable components. The Bch diet contained butter from Hacendado (Valencia, Spain), composed of a total content (%) of monounsaturated fatty acid (MUFA: 29% oleic acid, C18:1), saturated fatty acid (SAFA: 62% palmitic and stearic acid, C16:0 and C18:0), classified polyunsaturated fatty acids (PUFA: 4% C16:0), short and medium chain fatty acids (C4–C14) and α-tocopherol (0.0004 g/100 g). The VOO and Bch diets were isocaloric among themselves (VOO: 1848 KJ/100 g vs. Bch: 1827 KJ/100 g), but hypercaloric with respect to the S diet (1392 KJ/100 g), as described in previous works laboratory studies where only the Bch diet shows alterations in lipid metabolism, increased adipocytokine, leptin, increased lipemia and body weight as previously described [23,24,25]. At the end of 24 weeks, the rats were sacrificed under deep anesthesia with Equitensin (2 mL/kg body weight) and perfused with saline solution (0.9% NaCl) through the left ventricle. The brain and left adrenal gland were removed, rapidly freezing in liquid nitrogen to separate the hypothalamus and the anterior lobe region of the pituitary (adenohypophysis), as previously described [25,52] and using the Paxinos and Watson brain atlas as a reference [53]. The samples were stored at −80 °C until use.

### 2.2. Assay of Aminopeptidase Activities

Homogenization of the samples was necessary to separate them into two fractions, soluble (sol) and bound to the membrane (mb). A first homogenization of the samples in 10 volumes of 10 mM Tris-HCl buffer (pH 7.4) and their subsequent ultracentrifugation at 100,000× *g* for 30 min at 4 °C allowed obtaining a supernatant containing the soluble fraction, a fraction that it will serve for the measurement in triplicate of the enzymatic activities and the protein content. The solubilization of the pellet with the previous homogenization buffer to which 1% Triton-X-100 detergent is added will allow the extraction of proteins from the membrane after ultracentrifuge again at 100,000× *g* for 30 min at 4 °C. The supernatants were kept at least 4 h at 4 °C under agitation with SM-2 biobeads (100 mg/mL) (Bio-Rad, Richmond, VA, USA) to remove the detergent and solubilize the membrane proteins. The resulting samples corresponding to the membrane-bound fraction were used to again measure enzyme activities and protein content in triplicate.

The enzymatic activities of sol/mb fractions of angiotensinases (AlaAP, ArgAP, AspAP, CysAP and GluAP), APs of eating behavior (DPP-IV and pGluAP), stress level regulator’s APs (TyrAP) and biomarker Proline-iminopeptidase (PIP) were assesed by a fluorometric method using amino-acyl-β-naphthylamides (aa-β-NA) as substrates according to the methods of different authors [54,55,56] modified by Prieto and Ramírez [57,58]: L-Ala-, L-Arg-, L-Asp-, L-Cys- or L-Glu-β-NA for angiotensinases; L-Gly-Pro-β-NA for DPP-IV: L-pGlu-β-NA for pGluAP; L-Pro-β-NA for PIP; and L-Tyr-β-NA for TyrAP, respectively. Inside 96-well black plates, 10 µL/well of the supernatant sample (sol or mb) was pipetted along with 100 µL/well of substrate solutions, accompanied by an incubation at 37 °C for 30 min. Reactions are automatically stopped after 30 min of incubation by the addition of 100 µL/well of 0.1 M acetate buffer (pH 4.2). The enzymatic activity releases β-NA easily quantifiable by fluorimetry at emission wavelength of 412 nm and excitation wavelength of 345 nm. The activities of each tissue fraction were expressed as pmol of L-aa-β-NA hydrolysates per minute and per mg of protein (pmol aa-β-NA/min/mg prot). All chemical products were supplied by Sigma-Aldrich (St. Louis, MO, USA). PIP (EC 3.4.11.5) is considered a local biomarker [59,60,61,62] capable of hydrolyzing prolyl NH2-terminal residues of various peptides [59], however, its functional relationship in the brain to RAS changes by different HFDs is unknown.

### 2.3. Protein Measurement

Proteins were quantified by the Bradford method [63], using a bovine serum albumin prediluted protein concentration standard (1.0 mg/mL) (BSA; Sigma-Aldrich, St. Louis, MO, USA) dissolved in homogenization buffer.

### 2.4. Statistic Analysis

Statistical analysis was determined with one-way ANOVA and identifying multiple comparisons with Tukey’s post-hoc test. When the normality test is not successful, it was verified with a Kruskal-Wallis one-way range analysis of variance. Pearson’s correlation coefficient helped establish the relationship between the AP activities of the HPA axis. Statistical estimates were made with Sigmaplot v11.0 software (Systat Software, Inc., San José, CA, USA). Statistically significant differences were considered when the *p* value was less than 0.05 (*p* < 0.05). Values are presented as mean ± standard error of the mean (SEM).

## 3. Results

The experimental diets showed significant changes in the proteolytic activities of the pituitary gland and the adrenal gland. In the hypothalamus, no significant changes were observed in all the activities measured after the administration of the HFDs (data in Appendix A).

### 3.1. Angiotensinase Activities

In summary, the Bch diet showed increases in the activities AlaAP (sol/mb), ArgAP (mb), and CysAP (sol) in the pituitary gland compared to the S and VOO diet. Interestingly, the VOO diet presented similar values to the S diet. In the adrenal gland, only the AspAP (sol) activity with the HFDs was significantly elevated compared to the S diet.

#### 3.1.1. Aspartyl-Aminopeptidase Activity

The significant conversion of Ang-I to Ang 2–10 was carried out only in the adrenal gland (Figure 2F). AspAP activity was not significant with any of the HFDs in the pituitary (Figure 2A), however, with the HFDs it showed high activity only in the soluble fraction of the adrenal gland (S: 18.19 ± 0.45; VOO: 28.33 ± 3.79; Bch: 27.23 ± 3.81; Figure 2F, left). With the Bch diet, AspAP activity was elevated in both fractions of the pituitary gland, but without being significant (Figure 2A).

#### 3.1.2. Glutamyl-Aminopeptidase Activity

The experimental diets did not show significant differences in the hydrolyzing activity of Ang-II (Figure 2B–G). Despite this, there is evidence indicating an increase in GluAP (mb) activity in the pituitary (Figure 2B, right).

#### 3.1.3. Alanyl-Aminopeptidase and Arginyl-Aminopeptidase Activities

The metabolism of Ang-III into Ang-IV was highlighted by AlaAP and ArgAP activities in the pituitary gland, but not in the adrenal tissue (Figure 2C,D,H,I). The Bch diet increased AlaAP activity with respect to the S and VOO diets in both fractions of the pituitary gland (soluble fraction, S: 2027.08 ± 180.69; VOO: 2361.64 ± 112.38; Bch: 2955.28 ± 75.31; membrane-bound fraction, S: 4120.29 ± 286.42; VOO: 4156.58 ± 227.06; Bch: 5553.27 ± 522.90; Figure 2C–H). Similar to AlaAP activity, an increase in ArgAP (mb) activity was also found against S and VOO diets of the pituitary (S: 2876.15 ± 299.72; VOO: 3023.28 ± 47.22; Bch: 3810.46 ± 213.85; Figure 2D, right). HFD did not modify AlaAP and ArgAP activities in the adrenal gland (Figure 2H,I), despite the existence of indications of an increase in ArgAP (sol) activity with the VOO diet (Figure 2I, left). On the other hand, the non-significant increases of the Bch diet in the activities AlaAP (mb) and ArgAP (sol and mb) of the adrenal gland are also evidenced to a greater extent (Figure 2H,I).

#### 3.1.4. Cystinyl-Aminopeptidase Activity

As a continuation of the previous metabolic pathway, only the Bch diet was able to regulate CysAP activity, which translates into an increase in AT4 receptor or CysAP (sol) activity with diet Bch versus S and VOO diets in the pituitary (S: 614.03 ± 37.24; VOO: 666.70 ± 32.79; Bch: 822.35 ± 48.02; Figure 2E, left). The adrenal gland did not present alterations in CysAP activity with HFDs (Figure 2J), despite showing signs of increased activity with the VOO and Bch diets in its soluble fraction, and only with the Bch diet in the membrane bound fraction.

#### 3.1.5. Interrelation and Intrarelation of Angiotensinase Activities in the Tissues That Form the Hypothalamic-Pituitary-Adrenal Axis

Although the hypothalamus did not show significant differences in angiotensinase activities, it showed significant correlations between many angiotensinase activities of the hypothalamus (Figure 3 right; for more details see group correlation in Appendix A). Furthermore, hypothalamus and adenohypophysis also showed interrelationships for ArgAP (sol) and CysAP (sol) activities (Figure 3, left; for more details see: Appendix A). A stratified correlation (Appendix A) showed that the Bch diet was the cause of the group difference in the CysAP activities found between the hypothalamus and the pituitary. However, with the ArgAP activity between these two organs, the stratified correlation only observed an association with diet S. In the downstream axis, the CysAP (sol) activity was again interrelated between the pituitary and the adrenal gland (Figure 3 left; for more details see: Appendix A). A stratified correlation indicated (Appendix A) that the Bch diet did not reach significance but presented a strong association between the CysAP activity of both organs.

### 3.2. Dipeptidyl Peptidase IV, Pyroglutamyl-Aminopeptidase and Tyrosyl-Aminopeptidase Activities

The Bch diet significantly increased DPP-IV (sol) activity in the pituitary (S: 532.96 ± 28.05 vs. Bch: 651.73 ± 17.93; Figure 4C, left), but also suggests a non-significant increase of DPP-IV (mb) (Figure 4C, right). In the adrenal gland, increases in DPP-IV activity were suggested with both HFDs, but without being significant (Figure 4F).

None of the components of the HPA axis showed changes in the pGluAP activities after the administration of the HFD diets (Figure 4A,D; Appendix A). Despite this, a non-significant increase was shown in the adrenal gland with the two experimental diets, but the membrane fraction showed the highest activity with the Bch diet (Figure 4D).

Only the VOO diet was able to show less significant values of TyrAP activity in pituitary gland with respect to the Bch diet (VOO: 2377.77 ± 203.59 vs. Bch: 3072.27 ± 113.57; Figure 4B, left). However, in a non-significant way, with the Bch diet an increase in TyrAP activity was observed in all fractions of the pituitary and adrenal gland (Figure 4E).

### 3.3. Interrelation between Stress Activities and Eating Behavior with RAS Regulatory Activities

The interconnection of RAS activation to induce the regulation of other pathways is evident in Figure 1. The fractions of each tissue demonstrated the existence of correlations between the activities of energy metabolism associated with stress and eating behavior (DPP-IV and pGluAP) and of the behavior by enkephalinergic activity (TyrAP) with angiotensinase activities (Appendix A). In the hypothalamus, the interrelation between DPP-IV with pGluAP and TyrAP activities occurred only in the soluble fraction (Appendix A).

In the pituitary, the interrelationships between DPP-IV with pGluAP and TyrAP activities were significant in the soluble and membrane-bound fractions, but the interrelationship between the pGluAP and TyrAP activities were only significant in the soluble fraction (Appendix A).

In the adrenal gland, interrelationships between DPP-IV activities with pGluAP and TyrAP occurred in both the soluble and membrane-bound fractions. However, the correlation between the pGluAP with TyrAP activities were only significant in the membrane-bound fraction (Appendix A).

### 3.4. Proline-Iminopeptidase Activity like Neuromarker in Hypothalamus and Pituitary

The biomarker PIP (sol) showed in the pituitary a significant increase in activity with the Bch diet (S: 430.85 ± 15.78 vs. Bch: 520.04 ± 19.99; Figure 5, left; for more details: Appendix A). In addition, PIP (sol/mb) of the pituitary show significant correlations with angiotensinase activities and regulators of stress and behavior (Appendix A). An interesting data is found in the soluble fraction of the pituitary, where the PIP activity presented the highest correlations in the following order with DPP-IV > AlaAP > pGluAP activities. In the membrane-bound fraction of the pituitary, of all the most significant correlations, the highest appeared in the following order with CysAP > pGluAP > TyrAP.

The hypothalamus, despite having non-significant activities, presented correlations very similar to that of the pituitary (Appendix A).

## 4. Discussion

Chronic exposure to diets high in SAFA is known to act as a stressor that triggers stress due to hyperactivation of the HPA axis and associates it with a dominant increase in RAS and hypertension [9]. This work tries to understand how different types of HFDs (SAFA or MUFA) can intervene in the PA activities that regulate the RAS, stress and eating behavior, as well as to verify the existence of association between these activities. Furthermore, another interesting perspective was to check whether the cardiovascular benefits of olive oil have an implication on the mechanisms that regulate the aforementioned PA activities and their association between them. The administration of the Bch diet to the animals activated the non-classical RAS response in the anterior pituitary with the activities AlaAP (sol/mb), ArgAP (mb), CysAP (sol), in relation to the S and VOO diets. This central mechanism would explain in our experiments the increase in systolic BP (SBP) together with the changes in the angiotensinase activities of the kidney that we recently described, being the GluAP (sol) and CysAP (sol) activities of the renal medulla very high with the Bch diet compared to S and VOO diets [22].

Our results differ with those of other works, where the CysAP activity is not significant and the AlaAP activity is opposite in the frontal cortex of animals fed diets with a higher proportion of MUFA (olive oil or Iberian lard) or SAFA (coconut), However, diets with similar proportions of MUFA/PUFA and low proportion of SAFA (fish or sesame oil) presented lower values of AlaAP activity [28] that would correspond more precisely to the membrane-bound fraction [64].

A recent study in our laboratory in mice shows that with a diet of extra virgin olive oil (EVOO) the hypothalamic concentrations of Ang-II, Ang-III and Ang-IV are reduced, and neither the activity nor the gene expression change GluAP in hypothalamus [25]. In addition, a diet with butter increases the gene expression of GluAP and reduces its activity in the hypothalamus [25]. However, in our work we did not observe significant changes in any hypothalamic AP activity after HFD administration.

In the adrenal gland, a high response of AspAP (sol) activity was only observed with both HFDs Although the RAS axis regulates BP and acts on energy metabolism, the impact of HFDs on angiotensinases in the adrenal gland remains unknown. More studies would be needed to clarify the role of fatty acids on angiotensinase activities in the adrenal gland. In spite of everything, remarkable inter- and intra-correlations of angiotensinase activities were found between the components of the HPA axis, from which it was shown that separate diets have the ability to regulate APs activities.

Excess lipids in the diet alter the function of the HPA axis and promote anxiety-like behavior, however it is not entirely clear if these changes are based on high body weight and if these effects are specific to some type of fatty acid from the diet [37]. Other works contradict each other, where HFD would present anxiolytic properties and would be related to changes in HPA function [65]. There is evidence that prolonged high exposure to HFD causes leptin resistance and marked neuroendocrine impairment in central feedback loops of the HPA-norepinephrine stress axis [6]. A long-term SAFA diet intake, without increasing body weight, increases cortisol levels and modulates the central feedback processes of the HPA axis, without affecting anxiety-like behavior [37]. This suggests that the anxiogenic effects of prolonged HFD feeding may depend on a more pronounced metabolic dysfunction [37]. Due to this, this work focused on these aspects through APs activities that associate the regulation of RAS and metabolism associated with stress and behavior in the HPA axis.

DPP-IV, in addition to inactivating incretin hormones, mediates the degradation of many chemokines and neuropeptides. Decreased DPP-IV activity has been reported in sera from depressed patients [44]. However, increased DPP-IV activity is also associated with the prevalence of HFD-induced obesity, depression, and cognitive impairment in metabolic diseases in adults [66,67,68,69]. HFD induces regulation of DPP-IV substrate (NPY) in hypothalamic neurons [70], however, our results did not appreciate changes in DPP-IV activity in the hypothalamus, but a high DPP-IV activity in the pituitary with the Bch diet. A recent study of depression in mice shows that the impact of HFD lengthened behavior, as a consequence of a decrease in the expression of NPY in the hypothalamus and pituitary, while the levels of NPY and the DPP-IV activity increased in plasma [71].

Ang-II induces the secretion and release of catecholamines and NPY to a greater extent through stimulation of the AT1 receptor in primary cultures of human adrenal chromaffin cells [72]. HFD feeding affects catecholamine and NPY biosynthesis in the adrenal medulla in sympathetic response to stress along with increased BP, heart rate (HR), and delayed cardiovascular recovery after stress [73]. In addition, HFD increases the protein expression of the Ang-II receptor type 1 (AT1) in the hypothalamus, and also increases the levels of adrenomedullary tyrosine hydroxylase and NPY [73]. Our results did not show DPP-IV and TyrAP activity in the adrenal gland with HFD, but, as we have explained previously, an increase in DPP-IV activity was observed in the soluble fraction of the pituitary with the Bch diet and a higher TyrAP activity with the Bch diet than with the VOO diet in the soluble fraction of the pituitary. The intake behavior of the DPP-IV and TyrAP activities in the pituitary could be linked to previous results where, despite reducing the dietary intake with both HFDs, a significant increase in body weight was observed with the Bch diet and maintenance of weight with the VOO diet [23].

The significant difference between HFD diets in the adrenal gland would indicate that with the VOO diet it would have the lowest release of tyrosine residues and, with it, a reduction in the formation of catecholamines that would supposedly associate it with low levels of stress. In other words, the reduced TyrAP activity in the VOO diet would imply low levels of this enzyme which, together with a lower response in the induction of catecholamines, would exert a less analgesic role in the adenohypophysis due to not presenting disorders. The Bch diet showed a considerable non-significant increase in TyrAP activity in the pituitary and adrenal glands. Stress induces the activation of brain TyrAP [49] and a diet with virgin olive oil is capable of reducing the TyrAP activity in the soluble and membrane-bound fractions of the renal medulla [22].

As the highest stress on the APs regulatory activities of RAS and NPY hydrolysis occurred with the Bch diet in the soluble fraction of the pituitary sample, the marker of PIP functionality in the pituitary was subsequently verified. According to the previous results, the Bch diet also marked a significant increase in PIP activity in the soluble fraction of the pituitary. Possibly the high activity of this marker is the reflection of a significant correlation with APs activities regulating stress metabolism (DPP-IV) and RAS (AlaAP).

Activation of the RAS, or Ang-II itself, also seems to be involved in the effects of TRH on central cardiovascular regulation, being key in the etiology of hypertension [74]. The activation of AT1 receptors of the pituitary Ang-II and TRH stimulates the production of inositol phosphate, regulates the mobilization of intracellular Ca^2+^ and induces the secretion of adenohypophyseal hormones [75,76,77]. pGluAP has TRH as a substrate and produces the antioxidant and anti-inflammatory histidyl-proline diketopiperazine (cyclo [His-Pro]; cHP) as a product [78,79,80], but both exert effects on the CNS that include excitement and depression [47,81,82]. cHP has the ability to regulate central and local catecholamine levels [83,84], exert a neuroprotective role [80,85,86], stimulate neurogenesis in hippocampus [87] and treat metabolic diseases [88]. The metabolites of pGluAP have the ability to bind specifically to the adrenal glands and liver [23,89] and regulate catecholamine secretion [85]. Regulation of pGluAP appears to play a role in CNS stress and memory [83]. Previous studies in our laboratory identified that HFD modifies the local activity of pGluAP [24,49], however, our results with HFD did not alter pGluAP activity in the HPA axis, suggesting that it has no metabolic participation in cardiovascular regulation and energy homeostasis under stress conditions. It seems that imbalances in the HPA axes are accompanied by disorders in the hypothalamic-pituitary-thyroid (HPT) axis, both of which are frequently associated with CNS disorders, but the role of biological dysregulations of pituitary secretions on their target organs remains unclear [90,91].

This work showed as a novelty that olive oil can affect the regulation of blood pressure of HPA axis, protecting from an overactivation of the non-classic RAS, followed by a benefit on the enzymes that participate in dietary and stress control. However, there are limitations in our study that make it difficult to translate from animals to humans. Specifically, there is a lot of variability in the results of these experiments depending on the type of animal and proportions of the diet used. In addition, these results include a study started in a small number of male animals (*n* = 5–6/group) of advanced age (6 months of age), of which it would be necessary to increase the number of animals to reduce variability and include the female gender as a mixed study.

## 5. Conclusions

This study implies a participation of dietary fat on central and local APs that regulate RAS and stress components (Figure 6). The greatest involvement of angiotensinase activities occurs in the pituitary, showing an activation of the RAS pathway with the Bch diet; however, the VOO diet seems to go against the results with the Bch diet, presenting significantly lower values. In the adrenal gland, only its soluble fraction showed an increase in AspAP activity, suggesting a greater conversion of Ang-I to Ang 2–10. The pituitary also presented changes in the activities involved in stress behavior, where the Bch diet implied an increase in DPP-IV activities linked to an increase in the biomarker PIP. On the other hand, TyrAP (sol) activity in pituitary showed with VOO diet lower values of TyrAP activity with respect to the Bch diet, implying a decrease in sympathetic activity as a reflection of a lower angiotensinase activity. A future study of the possible interaction between the hypothalamic-pituitary-adrenal/thyroid axes associated with HFDs would allow to further narrow the relationship between cardiovascular homeostasis and stress energy homeostasis. Given that the axis of stress induced by HFD is closely regulated by leptin [6], it would be necessary to know how the saponifiable and minor components of virgin olive oil (in the short and long term) in adipose tissue influence dopaminergic, noradrenergic, thyroid (HPT) and adrenal (HPA) activities. Despite olive oil being a major benefactor in RAS regulation, its role in stress regulation remains unknown.

## Figures and Tables

**Figure 1 nutrients-13-03939-f001:**
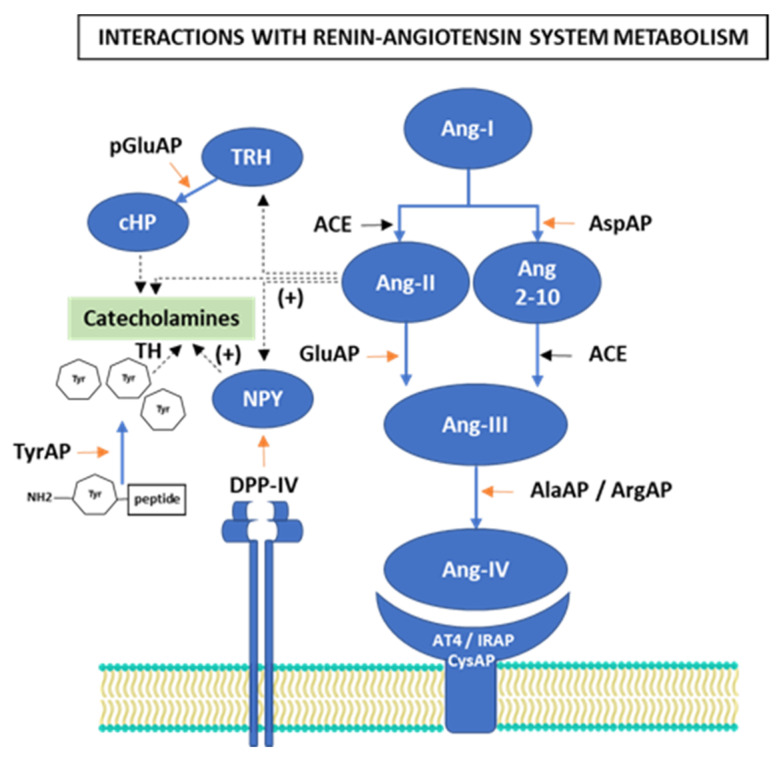
Downstream summary diagram of the classical and non-classical pathway of the renin-angiotensin system (RAS) and its interaction with stimulatory-hydrolyzing pathways of neuropeptide Y (NPY) and catecholamines. ACE: angiotensin converting enzyme AlaAP: alanyl-aminopeptidase; Ang 2-10: Angiotensin 2-10; Ang-I: angiotensin I; Ang-II: angiotensin II; Ang-III: angiotensin III; Ang-IV: angiotensin IV; ArgAP: arginyl-aminopeptidase; AspAP: aspartyl-aminopeptidase; cHP: histidyl-proline diketopiperazine; DPP-IV: dipeptidyl peptidase IV; GluAP: glutamyl-aminopeptidase; AT4/IRAP/CysAP: angiotensin IV receptor/insulin-regulated aminopeptidase/cystinyl-aminopeptidase; NH2: amino terminal; NPY: neuropeptide Y; pGluAP: pyroglutamyl-aminopeptidase; TH: tyrosine hydroxylase; TRH: Thyrotropin-releasing hormone; Tyr: tyrosine; TyrAP: tyrosyl-aminopeptidase.

**Figure 2 nutrients-13-03939-f002:**
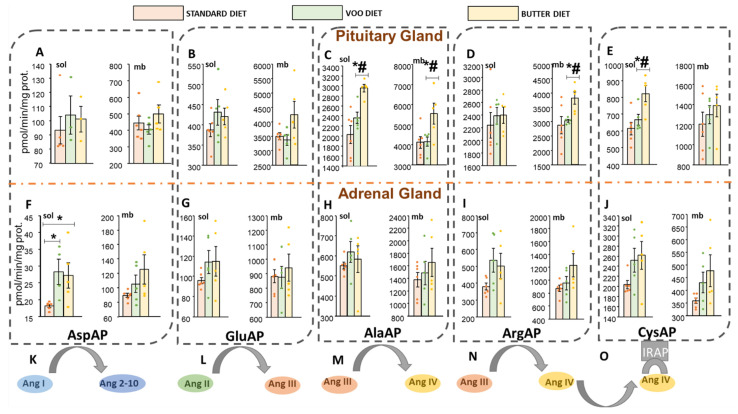
Angiotensin metabolism by angiotensinase activities. (**K**) Scheme of the metabolic conversion of angiotensin I (Ang-I) to angiotensin 2–10 (Ang 2–10) by the action of Aspartyl-aminopeptidase (AspAP). AspAP activity in the soluble (sol) and membrane-bound (mb) fractions of the (**A**) pituitary and (**F**) adrenal glands. (**L**) Scheme of the metabolic conversion of angiotensin II (Ang-II) to angiotensin III (Ang-III) by the action of Glutamyl-aminopeptidase (GluAP). GluAP activity in the sol and mb fractions of the (**B**) pituitary and (**G**) adrenal glands. (**M**,**N**) Scheme of the metabolic conversion of angiotensin III (Ang-III) to angiotensin IV (Ang-IV) by the action of Alanyl-aminopeptidase (AlaAP) and arginyl-aminopeptidase (ArgAP). AlaAP and ArgAP activities in the sol and mb fractions of the (**C**,**D**) pituitary and (**H**,**I**) adrenal glands. (**L**) Cystinyl-aminopeptidase activity (CysAP), also called insulin-related aminopeptidase (IRAP). Scheme of the binding of angiotensin IV (Ang-IV) to its ang-IV receptor (AT4) (**O**), which in turn is CysAP, with the ability to hydrolyze oxytocin and vasopressin. CysAP activity in the sol and mb fractions of the (**E**) pituitary and (**J**) adrenal glands. S: standard diet; VOO: diet enriched with virgin olive oil; Bch: diet enriched with diets and cholesterol. The activities were expressed as picomoles of β-naphthylamide per minute per milligram of protein (pmol/min/mg prot.). * *p*-value < 0.05, compared to standard diet (S). # *p*-value < 0.05, compared between HFD (VOO vs. Bch).

**Figure 3 nutrients-13-03939-f003:**
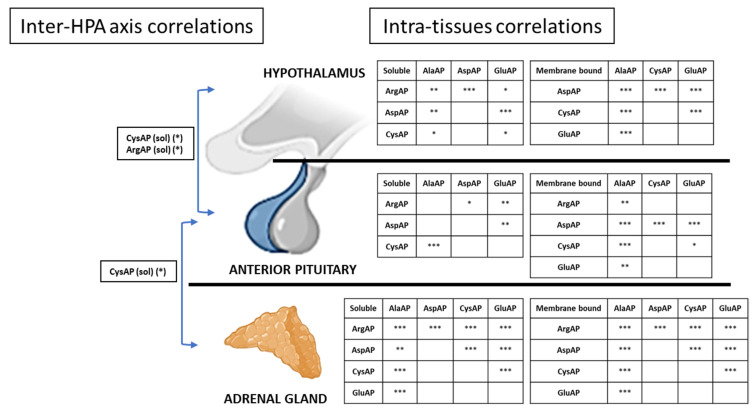
Correlation of the angiotensinase activities of the tissues (intra-tissues correlations, see Appendix A) and between the tissues (inter-HPA axis correlations, see Appendix A) that form the hypothalamic-pituitary-adrenal (HPA) axis. AlaAP: alanyl-aminopeptidase; ArgAP: arginyl-aminopeptidase; AspAP: aspartyl-aminopeptidase; CysAP: cystinyl-aminopeptidase; GluAP: glutamyl aminopeptidase; mb: membrane bound fraction; sol: soluble fraction. Asterisks represent *p*-value * < 0.05, ** < 0.01, *** < 0.001.

**Figure 4 nutrients-13-03939-f004:**
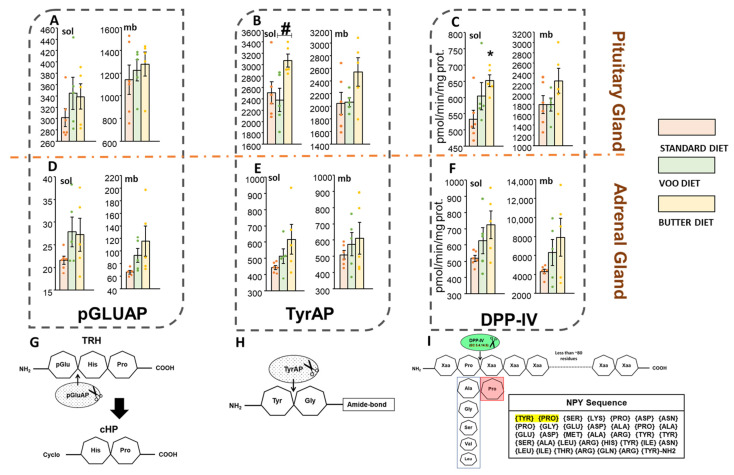
Metabolism of Dipeptidyl peptidase IV (DPP-IV), pyroglutamyl-aminopeptidase (pGluAP) and tyrosyl-aminopeptidase (TyrAP) activities. (**G**) Schematic of the metabolic conversion of thyrotropin-releasing hormone (TRH) to histidyl-proline diketopiperazine (cyclo [His-Pro]; CHP) by hydrolytic action of pGluAP activity. (**A**,**D**) pGluAP (sol and mb) activity of the pituitary and adrenal glands. (**H**) Schematic representation of TyrAP activity, which cleaves the Tyr-Gly amide-bond. (**B**,**E**) TyrAP (sol and mb) activity of the pituitary and adrenal glands. (**I**) Schematic representation of DPP-IV activity, that cleaves on the carboxy terminal (C-terminal) side of the proline residue. DPP-IV liberates a dipeptide from its substrates (below 80–100 residues) with proline (Pro) or alanine (Ala) at the second amino (NH2)-terminal position. It is remarked other preferred substrates, but with a slower cleavage rate depending on the residue that forms the second NH2-terminal position. DPP-IV cannot cleave substrates if the third residue is proline. The complete sequence of neuropeptide Y (NPY) appears in the inset, with a red arrowhead indicating the site of hydrolysis of DPP-IV. (**C**,**F**) DPP-IV activity in the soluble (sol) and membrane-bound (mb) fractions of the pituitary and adrenal glands. S: standard diet; VOO: diet enriched with virgin olive oil; Bch: diet enriched with butter and cholesterol. The activities were expressed as picomoles of β-naphthylamide per minute per milligram of protein (pmol/min/mg prot.). * *p*-value < 0.05, compared to standard diet (S). # *p*-value < 0.05, compared between HFDs (VOO vs. Bch).

**Figure 5 nutrients-13-03939-f005:**
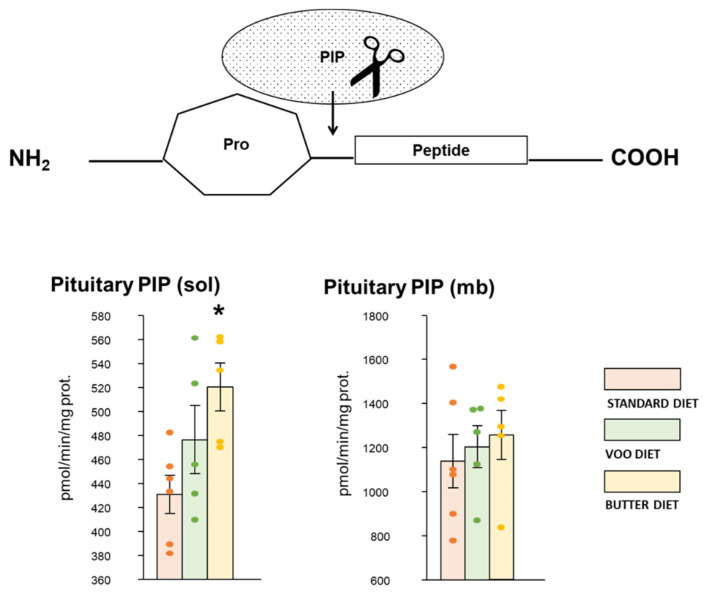
Proline-iminopeptidase (PIP) activity as neuromarker in pituitary, which cleaves the amino (NH2)-Proline (Pro) of peptide substrate. Mb: membrane-bound (mb) fraction; S: standard diet; Sol: soluble fraction; VOO: diet enriched with virgin olive oil; Bch: diet enriched with butter and cholesterol. The activities were expressed as picomoles of β-naphthylamide per minute per milligram of protein (pmol/min/mg prot.). * *p*-value < 0.05, compared to standard diet (S).

**Figure 6 nutrients-13-03939-f006:**
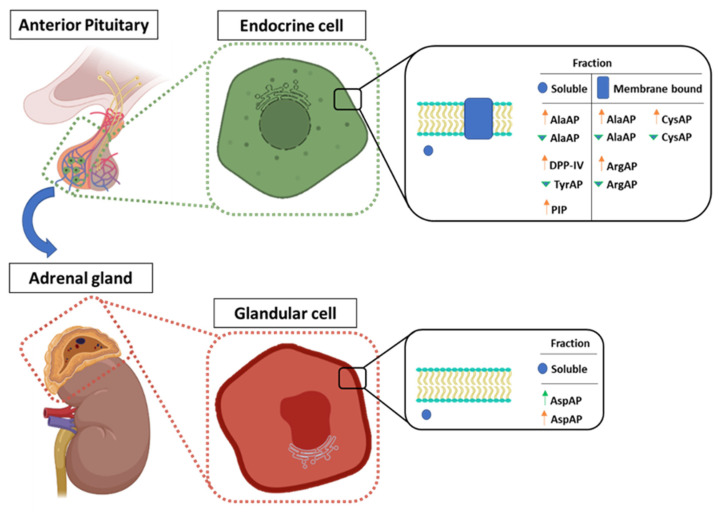
Summary diagram of the modified enzyme activities in the soluble and membrane-bound fractions of the anterior pituitary (adenohypophysis) and the adrenal gland. The green and orange arrows indicate diets with olive oil (VOO) and butter plus cholesterol (Bch), respectively. The arrow (↑) represents significant differences of a high-fat diet compared to the standard diet (S) (VOO or Bch vs. S). The arrowhead (▼) represents significant differences between high-fat diets (VOO vs. Bch).

## Data Availability

Not applicable.

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
