# Peer review of "The Type of Fat in the Diet Influences Regulatory Aminopeptidases of the Renin-Angiotensin System and Stress in the Hypothalamic-Pituitary-Adrenal Axis in Adult Wistar Rats"

_nutrients, 2021, doi:10.3390/nu13113939_

Round 1

Reviewer 1 Report

In this manuscript, Dominguez-Vias and colleagues aimed at understanding the effect of different types of fats in the regulation of the Renin-Angiotensin system in the HPA axis. To do so they subjected Wistar rats to different types of diets and measured the activity of different aminopeptidases in the hypothalamus, pituitary, and adrenal gland. This work is rather preliminary and has many concerns.

Major concern

- In this study, the authors have provided different types of diets for 24 weeks to rats. The authors mentioned that VOO and Bch diets are hypercaloric regarding the control diet. What is the effect of these diets on energy metabolism? Do these rats become obese/diabetic after being fed with VOO and Bch?

- The authors performed several correlation analyses in this study, and the conclusion drawn from these analyses is the main claim of the manuscript. In the first correlation analysis, the authors highlight a correlation between angiotensinase activities from the HPA axis. How is this related to the focus of the manuscript (the effect of fat on Renin-Angiotensin system)? In a second correlations analysis, authors are claiming a correlation between Renin-Angiotensin system and stress/eating behavior. In a similar manner, this is not related to the focus of the manuscript (the effect of fat on Renin-Angiotensin system). Moreover, I could not find any data where the authors have measured stress or eating behavior in these animals.

- The authors found that some aminopeptidase activities are modulated by the fat from Bch or VOO diets. What is the effect on the accumulation of angiotensin? In addition, are these diets modulating markers of the HPA axis (cortisol, ACHT, NPY…)? It could be interesting to know if the change in aminopeptidase activities is followed by changes in HPA axis markers.

Minor concerns

- The figures of the manuscript are way too small and it was very difficult to read them.

- The authors mentioned no effect on aminopeptidases activity in the hypothalamus (data not shown). The authors must show the data.

- In the discussion what mean “MR”?

Author Response

Authors’ Response to the Reviewer’s comments

Journal:               Nutrients

Title of Paper:     The Type of Fat in the Diet Influences Regulatory Aminopeptidases of the Renin-Angiotensin System and Stress in the Hypothalamic-Pituitary-Adrenal Axis in Adult Wistar rats

Authors:              Germán Domínguez-Vías, Ana Belén Segarra, Manuel Ramírez-Sánchez, Isabel Prieto

Date Sent:           27 October, 2021

Dear Reviewer,

We appreciate your time and efforts in reviewing our manuscript. We are delighted to answer your question.

Yours faithfully,

Germán Domínguez-Vías

Reviewer 1:

In this manuscript, Dominguez-Vias and colleagues aimed at understanding the effect of different types of fats in the regulation of the Renin-Angiotensin system in the HPA axis. To do so they subjected Wistar rats to different types of diets and measured the activity of different aminopeptidases in the hypothalamus, pituitary, and adrenal gland. This work is rather preliminary and has many concerns.

I appreciate the reviewer's comments and below I detail the answers to each of the questions.

Major concern

- In this study, the authors have provided different types of diets for 24 weeks to rats. The authors mentioned that VOO and Bch diets are hypercaloric regarding the control diet.

What is the effect of these diets on energy metabolism? Do these rats become obese/diabetic after being fed with VOO and Bch?

Our laboratory recently carried out this study with this type of diet. We report that the Bch diet significantly reduces the expression of genes related to fatty acid catabolism (LPL, CD36 and CPT-1) in liver and skeletal muscle. The inclusion of a VOO diet does not affect the mRNA expression of these genes in the liver, but in the muscle it does significantly affect the expression of the LPL and CD36 genes. Nor were differences found in the expression of the FAS gene between the three diets in liver and muscle, although in muscle the inclusion of olive oil presents higher values ​​without being significant (23).

In our diet models we have not found alterations in glucose homeostasis (24), however, we have published that the Bch diet significantly increases body weight and total visceral adipose tissue (23,25), producing high levels of postprandial leptin (23). With the Bch diet we have found an increase in plasma triglycerides, total cholesterol, LDL and VLDL; but at no time have we observed changes with the VOO diet in terms of body weight, total visceral fat, and expression of liver genes related to fatty acid metabolism. The inclusion of the VOO diet decreases leptin levels and increases fasting ghrelin levels, along with a lesser effect on the metabolism of fatty acids in the muscles and the liver and an adequate maintenance of body weight (23).

Cites 23, 24 and 25:

-             (23) Segarra, A.B.; Domínguez-Vías, G.; Redondo, J.; Martínez-Cañamero, M.; Ramírez-Sánchez, M.; Prieto, I. Hypothalamic Renin–Angiotensin System and Lipid Metabolism: Effects of Virgin Olive Oil versus Butter in the Diet. Nutrients 2021, 13, 480. https://doi.org/10.3390/nu13020480

-             (24) Domínguez-Vías, G.; Segarra, A.B.; Ramírez-Sánchez, M.; Prieto, I. The Role of High Fat Diets and Liver Peptidase Activity in the Development of Obesity and Insulin Resistance in Wistar Rats. Nutrients 2020, 12, 636. https://doi.org/10.3390/nu12030636

-             (25) Domínguez-Vías, G.; Segarra, A.B.; Martínez-Cañamero, M.; Ramírez-Sánchez, M.; Prieto, I. Influence of a Virgin Olive Oil versus Butter Plus Cholesterol-Enriched Diet on Testicular Enzymatic Activities in Adult Male Rats. Int. J. Mol. Sci. 2017, 18, 1701. https://doi.org/10.3390/ijms18081701

We have added the following paragraph in "material and method" (lines 151 -156):

“… The VOO and Bch diets were isocaloric among themselves (VOO: 1848 KJ/100 g vs. Bch: 1827 KJ/100 g), but hypercaloric with respect to the S diet (1392 KJ/100 g), as described in previous works laboratory studies where only the Bch diet shows alterations in lipid metabolism, increased adipocytokine, leptin, increased lipemia and body weight as previously described [23-25]. ".

- The authors performed several correlation analyses in this study, and the conclusion drawn from these analyses is the main claim of the manuscript. In the first correlation analysis, the authors highlight a correlation between angiotensinase activities from the HPA axis. How is this related to the focus of the manuscript (the effect of fat on Renin-Angiotensin system)?

The type of diet does seem to condition the correlations of aminopeptidase activities in different tissues analyzed. For example, in the case of ArgAP (sol) activity in the hypothalamus and pituitary, when the three groups of animals are included there is a level of significance, and stratified (analyzed separately) there is a correlation between the diet group S but not in the HFDs (data included for consultation in the supplementary material).

 In a second correlations analysis, authors are claiming a correlation between Renin-Angiotensin system and stress/eating behavior. In a similar manner, this is not related to the focus of the manuscript (the effect of fat on Renin-Angiotensin system).

It is likely that it was not clear in the writing of the manuscript. The correlations are between angiotensin activities of the RAS and other distinct aminopeptidase activities related to stress and eating behavior. The working objective (lines 107 – 109) and method description  have been rewritten to report that the correlations are between aminopeptidases with different functions.

Moreover, I could not find any data where the authors have measured stress or eating behavior in these animals.

It is known that physiological responses to stress are also a consequence of the consumption of a diet high in saturated fat (SAFA) and have been associated with deterioration of physical and mental health (Sivanathan et al., 2015). For this reason, the aminopeptidase activities TyrAP, pGluAP and DPP-IV were measured, because they are involved in the regulation of the level of stress (TyrAP), and of eating behavior associated with stress (pGluAP and DPP-IV). This aspect has been briefly redrafted in the introduction to avoid confussion and it is emphasized several times during the manuscript.

Cite: Sivanathan, S., Thavartnam, K., Arif, S., Elegino, T., & McGowan, P. O. (2015). Chronic high fat feeding increases anxiety-like behaviour and reduces transcript abundance of glucocorticoid signalling genes in the hippocampus of female rats. Behavioural brain research, 286, 265–270. https://doi.org/10.1016/j.bbr.2015.02.036

- The authors found that some aminopeptidase activities are modulated by the fat from Bch or VOO diets. What is the effect on the accumulation of angiotensin?

In this experiment, unfortunately, we did not measure angiotensin levels, but with a different animal model in another feeding work, carried out with the same types of diet, we verified that in the hypothalamus the two HFD diets decrease AngII and AngII, but AngIII only decreases with VOO (reference number 23).

In addition, are these diets modulating markers of the HPA axis (cortisol, ACHT, NPY…)? It could be interesting to know if the change in aminopeptidase activities is followed by changes in HPA axis markers.

The suggestion is very interesting, unfortunately in these samples we have not been able to obtain measurements of these parameters.

Minor concerns

- The figures of the manuscript are way too small and it was very difficult to read them.

The figures have been modified and/or redo for a better reading and understanding.

- The authors mentioned no effect on aminopeptidases activity in the hypothalamus (data not shown). The authors must show the data.

Sorry for the mistake of not indicating that those values were in supplemental material. I correct in the text and add that specification (lines 208 – 209).

- In the discussion what mean “MR”?

It is true, it is a mistake not to have described it. MR stands for "Mineralocorticoid Receptor". In the new rewrite that term is omitted for reasons of brevity in the text and to obviate explanations that are far from my purpose.

Reviewer 2 Report

The present study presents interesting data that associates different types of fat with changes in aminopeptidases enzymes in the brain regions that coordinate HPA axis responses. The authors have completed a great deal of work, and there is merit in the data collection and figures. However, there are flaws in the presentation of literature and overall writing style.

Introduction:

There is a lot of information presented in a manner which is too condensed. A more general introduction would be helpful for the reader, including some general background on the HPA axis, the RAS and how it relates to different fat diet etc. In its present form, the introduction is very difficult to read. Please consider structurally it more carefully so each paragraph is a subsection which helps provide the overall ‘story’ of your project rationale. Please don’t include any information which isn’t relevant to your experiments.

Materials and Methods:

  1. Please provide a reference/rationale for why 24 weeks was chosen for the diet intervention.
  2. Was there a reason the experiments started at 6 months of age?
  3. Please state if rats were group housed.

Results:

  1. Please provide body weight data over the course of the experiment and food intake data. It would be important to determine any differences in total food intake/body fat composition which could account for the effects seen.
  2. The hypothalamus is a large region. Please describe if you were aiming for the most HPA axis-relevant area of the hypothalamus – the paraventricular nucleus. If you weren’t selectively looking at the PVN this could account for the lack of statistical differences seen in the hypothalamus.
  3. Please make your graphs larger, in the current state it is difficult to see.

Discussion:

The discussion is not well organized and difficult to read in the present form. Consider starting the discussion with a sentence on the rationale of your work and then provide a summary of what you found. Consider having subsequent paragraphs each discussing a particular finding so the reader can easily follow along. Please don’t include any information which isn’t relevant to your findings.

Please invite a native English speaker to review this manuscript.

Author Response

Authors’ Response to the Reviewer’s comments

Journal:               Nutrients

Title of Paper:     The Type of Fat in the Diet Influences Regulatory Aminopeptidases of the Renin-Angiotensin System and Stress in the Hypothalamic-Pituitary-Adrenal Axis in Adult Wistar rats

Authors:              Germán Domínguez-Vías, Ana Belén Segarra, Manuel Ramírez-Sánchez, Isabel Prieto

Date Sent:           27 October, 2021

Dear Reviewer,

We appreciate your time and efforts in reviewing our manuscript. All the issues indicated in the comments from Reviewer have been addressed.

We have reduced and simplified the content of the manuscript. Along with simplifying and eliminating data irrelevant to our work, we have rewritten the material and method section to avoid similarities with previous work of ours. Reviewer 'suggestions, are marked in yellow in the text. Also at the request of the reviewers, we have modified and/or redone the figures so that they have greater visibility for the reader. I have also corrected a small error where it indicated that did not show data from the hypothalamus, when it actually did show all the results in supplemental material. Major changes, made according to the Reviewer 'suggestions, are marked in yellow in the text.

Yours faithfully,

Germán Domínguez-Vías

Reviewer 2:

The present study presents interesting data that associates different types of fat with changes in aminopeptidases enzymes in the brain regions that coordinate HPA axis responses. The authors have completed a great deal of work, and there is merit in the data collection and figures. However, there are flaws in the presentation of literature and overall writing style.

I thank the reviewer for his valuable comments. The manuscript has been rewritten to present the most useful and detailed information.

Introduction:

There is a lot of information presented in a manner which is too condensed. A more general introduction would be helpful for the reader, including some general background on the HPA axis, the RAS and how it relates to different fat diet etc. In its present form, the introduction is very difficult to read. Please consider structurally it more carefully so each paragraph is a subsection which helps provide the overall ‘story’ of your project rationale. Please don’t include any information which isn’t relevant to your experiments.

The introduction has been rewritten to facilitate reading and present only the information relevant to the work carried out.

Materials and Methods:

    Please provide a reference/rationale for why 24 weeks was chosen for the diet intervention.

  We have data that this type of diet causes metabolic alterations when administered for a long time, at least 24 weeks as described previously (references23 - 25). We include this information in material and method (In section: 2.1. Animals and Diets).

  Was there a reason the experiments started at 6 months of age?

   Yes, that age has been used in animals because we try to simulate the effect of a high-fat diet in adults. We have experience in analyzing the influence of different diets on different physiological functions at different ages, but we know that the use of adult male rats is a good model for the development of models that develop factors associated to the incidence of metabolic syndrome as in humans (references 22 - 24). That is, we justify this advanced age especially to analyze possible differences from 6 months to 12 months of age, due to its equivalence to 46-year-old humans (Sengupta, 2013).

Reference:

Sengupta, P. The Laboratory Rat: Relating Its Age With Human’s. Int. J. Prev. Med. 2013, 4, 624–630.

 Please state if rats were group housed.

Yes, the rats were separated by dietary groups. We include this information in material and method.

Results:

    Please provide body weight data over the course of the experiment and food intake data. It would be important to determine any differences in total food intake/body fat composition which could account for the effects seen.

With the Bch diet we have found an increase in plasma triglycerides, total cholesterol, LDL and VLDL; but at no time have we observed changes with the VOO diet in terms of body weight, total visceral fat, and expression of liver genes related to fatty acid metabolism. We have already verified and published these data (references number 23 and 24). The inclusion of the VOO diet decreases leptin levels and increases fasting ghrelin levels, along with a lesser effect on the metabolism of fatty acids in the muscles and the liver and an adequate maintenance of body weight (reference number 25).

    The hypothalamus is a large region. Please describe if you were aiming for the most HPA axis-relevant area of the hypothalamus – the paraventricular nucleus. If you weren’t selectively looking at the PVN this could account for the lack of statistical differences seen in the hypothalamus.

We have used the entire hypothalamus, but indeed if we had analyzed separate nuclei we would have located more precise differences. However, in another work carried out on another experimental model in our laboratory, we did obtain differences (reference number 25).

    Please make your graphs larger, in the current state it is difficult to see.

The graphics have been modified and/or redo for a better visualization.

Discussion:

The discussion is not well organized and difficult to read in the present form. Consider starting the discussion with a sentence on the rationale of your work and then provide a summary of what you found. Consider having subsequent paragraphs each discussing a particular finding so the reader can easily follow along. Please don’t include any information which isn’t relevant to your findings.

I appreciate the recommended guidelines. A restructuring of the manuscript has been carried out following their recommendations.

Please invite a native English speaker to review this manuscript.

Due to the urgency of the requested times to send the manuscript, we have not been able to send it to the translation service. If it is accepted, before its publication we promise to send it to the English language editing service of the MDPI publisher.

Round 2

Reviewer 1 Report

The authors have answered all of my comments. The manuscript has been substantially improved.